# The Role of Vascular Smooth Muscle Cells in Arterial Remodeling: Focus on Calcification-Related Processes

**DOI:** 10.3390/ijms20225694

**Published:** 2019-11-14

**Authors:** Armand Jaminon, Koen Reesink, Abraham Kroon, Leon Schurgers

**Affiliations:** 1Department of Biochemistry, Cardiovascular Research Institute Maastricht (CARIM), Maastricht University, 6229 ER Maastricht, The Netherlands; a.jaminon@maastrichtuniversity.nl; 2Department of Biomedical Engineering, Cardiovascular Research Institute Maastricht (CARIM), Maastricht University, 6229 ER Maastricht, The Netherlands; k.reesink@maastrichtuniversity.nl; 3Department of Internal Medicine, Maastricht University Medical Centre (MUMC+), 6229 HX Maastricht, The Netherlands; aa.kroon@mumc.nl

**Keywords:** arterial remodeling, vascular smooth muscle cell, vascular calcification, vascular stiffness, hypertension, phenotype switching

## Abstract

Arterial remodeling refers to the structural and functional changes of the vessel wall that occur in response to disease, injury, or aging. Vascular smooth muscle cells (VSMC) play a pivotal role in regulating the remodeling processes of the vessel wall. Phenotypic switching of VSMC involves oxidative stress-induced extracellular vesicle release, driving calcification processes. The VSMC phenotype is relevant to plaque initiation, development and stability, whereas, in the media, the VSMC phenotype is important in maintaining tissue elasticity, wall stress homeostasis and vessel stiffness. Clinically, assessment of arterial remodeling is a challenge; particularly distinguishing intimal and medial involvement, and their contributions to vessel wall remodeling. The limitations pertain to imaging resolution and sensitivity, so methodological development is focused on improving those. Moreover, the integration of data across the microscopic (i.e., cell-tissue) and macroscopic (i.e., vessel-system) scale for correct interpretation is innately challenging, because of the multiple biophysical and biochemical factors involved. In the present review, we describe the arterial remodeling processes that govern arterial stiffening, atherosclerosis and calcification, with a particular focus on VSMC phenotypic switching. Additionally, we review clinically applicable methodologies to assess arterial remodeling and the latest developments in these, seeking to unravel the ubiquitous corroborator of vascular pathology that calcification appears to be.

## 1. Introduction

The vessel wall consists of multiple cells, including endothelial cells (ECs), vascular smooth muscle cells (VSMCs), fibroblasts and pericytes, and structural extracellular matrix (ECM) components, such as elastin and collagen [1]. ECs form the innermost layer, being the barrier between the blood-carrying lumen and vessel wall. In the vessel wall, ECs are surrounded by the tunica media, consisting almost entirely of circumferentially oriented VSMCs. Medial VSMCs, facilitating vessel dilation and constriction, are surrounded by interconnected elastic lamellae [2]. The outer layer of the vessel wall is the adventitia and contains ECM, maintaining structural integrity under peak mechanical load, and fibroblasts and progenitor (mesenchymal stem cell-like, MSC-like) cells. The interplay of all components of the vessel wall is needed to preserve vascular health [3,4,5].

The vessel wall is continuously exposed to biomechanical and biochemical stressors that elicit functional and adaptative responses. For instance, when blood flow is acutely increased, the increased wall shear stress (or friction) is sensed by ECs, which release relaxants. These relaxants (e.g., nitric oxide, NO) lower the active tone of VSMCs, leading to (flow-mediated) vessel dilation to counteract the initial increase in wall shear stress [6]. Long-term changes in wall shear stress provoke diameter adaptation [7]. Wall thickness changes are seen in response to increased wall stress (or wall tension), due to high blood pressure [8]. Such arterial remodeling responses shape early development of the macro-circulation, but are also affected by normal aging processes [9]. A multitude of biological processes contribute to pathological vascular remodeling, such as inflammation, oxidative stress, lipid accumulation, and degradation of the ECM [10,11]. Due to the heterogeneity of vascular remodeling, it is difficult to pin-point single biological processes responsible for vascular disease. Lessons from genetic disorders such as pseudoxanthoma elasticum (PXE), Marfan’s syndrome and Keutel syndrome, have addressed the complexity of arterial remodeling [12]. A better understanding of the complexity of arterial remodeling will help unravel its role as a cause or consequence of cardiovascular disease (CVD).

VSMCs in the medial layer of the vessel wall play a key role in arterial remodeling. VSMCs are the most abundant cell type in the arterial vessel wall and are pivotal in maintaining vessel structure and function [13]. VSMCs are considered heterogeneous and display a high degree of plasticity [5]. Under physiological conditions, VSMCs have a contractile phenotype which facilitates the contraction and dilation of the vasculature, which, in smaller resistance arteries, is key to the regulation of blood flow. Upon biological stress signals or vascular injury, VSMCs respond by losing contractility markers and differentiate towards a synthetic VSMC phenotype. Synthetic VSMCs subsequently express proteins involved in proliferation and migration [14]. Contractile VSMCs may adapt to stress by differentiating towards synthetic VSMCs which are able to newly synthesize ECM components such as collagen. To remodel the ECM, synthetic VSMCs produce metalloproteinases (MMPs), such as collagenases and elastases, that allow them to migrate to sites of injury [15,16,17]. VSMC-mediated remodeling of ECM within the vessel wall may result in increased arterial stiffness [13,18], contributing to systolic hypertension and altered hemodynamic conditions in end-organs such as the brain, kidneys and heart [19].

VSMCs are involved in many vascular diseases, such as atherosclerosis and aneurysm formation. In all these vascular pathologies, vascular calcification is involved [5,20,21]. Vascular calcification is defined as the deposition of calcium crystals within the vessel wall, initiated as microcalcification and propagating towards macrocalcifications [22], and, eventually, encroaching entire segments of the vasculature [23]. Vascular calcification is an active process with a key role for VSMCs, including apoptosis [24], osteochondrogenic transdifferentiation [25], extracellular vesicle release [2], calcium overload [26], and cellular senescence [27] (Figure 1). The aim of the present review is to provide an overview of the pathways underlying arterial remodeling, with a particular focus on the role of VSMCs and calcification-related processes.

## 2. Biology of Vascular Remodeling

### 2.1. Arterial Remodeling

Arterial remodeling reflects the adaptation of the vessel wall to biochemical and biomechanical stimuli [9,12]. In the vasculature, two types of remodeling can be distinguished: outward and inward remodeling, with respective hypertrophy (thickening) or hypotrophy (thinning) of the vessel wall. Large conduit vessels do not have the ability to constrict in response to stress, and therefore show hypertrophic remodeling. Atherosclerosis is characterized by an increase in vessel diameter, with the thickening of both the media and intima, and classically termed outward hypertrophic remodeling [28]. Aneurysm formation is characterized by an increase in vessel diameter, with a thinning of the vessel wall, and termed outward hypothropic remodeling [29] (Figure 2). Inward remodeling is less frequently seen and is observed in more muscular peripheral arteries, probably reflecting the sustained vasoconstriction of vessels [30].

Under low laminar flow, platelet derived growth factors (PDGF) and transforming growth factor-β (TGF-β) promote inward remodeling by increasing VSMC proliferation and collagen deposition [31]. Inward remodeling also contributes to atherosclerosis, but does not lead to an increased vessel size (Figure 2). Atherosclerosis development involves many cellular processes, like the recruitment of inflammatory cells by chemotaxis and infiltration [32].

VSMCs are connected to a fenestrated network of elastin and collagen fibers. The capacity of the vessel wall to elastically distend is important to accommodate the volume ejected with each heartbeat and to limit peripheral pressure pulsations. The active tone and spatial arrangement of VSMCs may influence the mechanical load on the ECM components and, therefore, modulate vessel diameter and stiffness [18]. Chronic exposure to high blood pressure increases tensile stress [9] to which VSMCs respond by proliferation, resulting in hyperplasia and thickening of the vascular wall. Concomitant activity of MMPs facilitates the structural breakdown of elastin ECM, and synthetic VSMCs produce collagen ECM, attempting to preserve stiffness homeostasis [33]. High blood pressure thus aggravates age-related stiffening of arteries [34,35,36]. Additional ECM disturbances, such as the presence of calcium crystals, have a further impact on the stiffening of the medial layer [37,38].

During aging, it is generally accepted that the number of cells in the vasculature decreases, although the causes of this finding remain to be established [34]. It has been hypothesized that VSMCs become senescent and that cell division rates decrease [39]. The recent literature ignores vessel wall cellularity and often refers to cellular processes, such as apoptosis, inflammation, calcification and epigenetic effects, all playing a part in vessel wall aging [20]. Additionally, with aging, collagen content in major arterial vasculature increases, whereas elastin content decreases and the number of VSMCs declines [40]. As a consequence, remaining VSMCs are embedded in a collagen-enriched ECM with fewer cellular focal adhesions [41,42,43]. Within arterial vessels, differences exist in the content of elastin to VSMC ratios [44]. Large arteries close to the heart contain more elastin and are therefore called “elastic” arteries. It is particularly elastic arteries that stiffen with age [45]. Large artery stiffening results in decreased arterial compliance, especially in those aged over 60 years [19,46,47]. Peripheral vessels contain more VSMCs relative to elastin and are termed “muscular” arteries. In muscular arteries, the relative elastin content increases with age, most likely caused by the decline in the number of VSMCs and decreased collagen content [44]. It should be noted that the absolute amount of ECM proteins in the vasculature decreases with age, but that fat and extracellular material, such as calcium crystals, increase [48]. Taken together, the number of VSMCs within the vasculature strongly correlates with vascular stiffening and the arterial remodeling processes [34].

Endothelial function plays an important role in arterial remodeling. Blood flow and normal wall shear stress stimulate ECs to produce NO (Figure 3). NO induces the relaxation of local VSMCs, which leads to dilation of the vessel wall. Endothelium-dependent vasodilation decreases with age, which appears to be associated (perhaps causally or bi-directionally) with a quiescent state of VSMCs and is clearly implicated to be associated with hypertension and CVD [49,50]. Under pathological conditions, ECs are known to produce cytokines and growth factors [51], which induces VSMCs phenotype switching from contractile to a more synthetic phenotype. An increased wall shear stress increases endothelial derived NO release, further decreasing VSMC proliferation and increasing VSMC apoptosis [52], resulting in outward remodeling and stiffened vasculature, due to the formation of the ECM matrix by remaining synthetic VSMCs [34,53].

#### 2.1.1. Inflammation and Arterial Remodeling

Inflammatory processes play a significant role in arterial remodeling and are linked to the development of atherosclerosis. Chronic low-grade inflammation is a key driver of arterial aging. VSMC phenotypic switching and ECM biochemical changes activate inflammatory pathways and oxidative stress signaling [54]. Cytokines determine processes such as endothelial dysfunction, renin angiotensin activation and metalloproteinases release [55,56]. VSMCs are involved in local vessel wall inflammatory functions.

##### Proliferation

VSMCs have low turnover rates in healthy vessels, but increase proliferation upon vascular injury to initiate repair [57]. Increased expression of aging genes p16^INK4a^ and p21 [58] have been linked to VSMC senescence and are present in atherosclerotic plaques, indicating lower proliferation and, thus, decreased repair capacity [59].

##### Cell Death

Cell death is linked to vascular disease, but absolute rates are difficult to determine. VSMC apoptosis in atherosclerosis has been linked to plaque vulnerability features [60], and chronic VSMC apoptosis has been shown to promote atherogenesis and plaque progression [61]. In vascular aging and medial remodeling, VSMCs cell death does not cause significant inflammation, due to the efficient IL-1*β*-mediated clearance of apoptotic bodies by VSMCs [62].

##### Platelets and Extracellular Vesicles (EVs)

Platelets have shown to affect VSMC inflammatory phenotypes and injury responses [63]. Additionally, platelet EVs induce an inflammatory response in VSMC in vitro [64]. Platelet factor 4 (PF4) has a central role in the stimulation of VSMC-mediated cytokine release, which is primarily affected by increased krüppel-like factor 4 (KLF-4) transcription. Experimental atherosclerosis models indicated that PF4 is proatherogenic and it has been found to penetrate deep into the vessel wall, underlining the importance of platelets aside from their effect on the endothelium [65].

### 2.2. The Role of VSMC Plasticity in Vascular Remodeling

Under physiological conditions, VSMCs display a contractile phenotype, in which markers such as alpha-smooth muscle actin (α-SMA), smooth muscle 22-alpha (SM22-α), smoothelin A/B (smtn), smooth muscle-myosin heavy chain (MYC11) and Calponin-1 (CNN-1) are highly expressed [14,66,67] (Table 1). VSMCs are heterogeneous with regard to their expression, are derived from different embryonic origins and have a strong genetic component causing VSMC diversity. Numerous in vivo studies have shown the existence of diverse VSMC populations within the same artery in rats [68], pigs [69] and humans [70]. We refer to VSMC plasticity here as the capability of VSMCs to switch phenotype. VSMC phenotypes are changed by environmental cues, such as soluble biochemical compounds, ECM proteins and biophysical conditions [33,71] (Figure 1). VSMCs exhibit a whole array of phenotypes, ranging from contractile–quiescent to migratory–proliferative–synthetic and osteogenic-, macrophage- or MSC-like [72,73,74] (Table 1). In the following subsections, we will briefly describe the most relevant factors that affect VSMC phenotype (Figure 1 and Table 2).

#### 2.2.1. Biochemical Compounds

PDGF is an important signaling molecule in the initial phase of VSMC differentiation. During vascular development, PDGF causes mesenchymal cell recruitment and subsequent proliferation [75]. PDGF is known to induce VSMC differentiation towards a synthetic phenotype, as it downregulates the contractile marker αSMA in aortic VSMCs [76]. Additionally, PDGF increases proliferation and migration in pig and human coronary VSMCs [69,77,78]. Moreover, in vivo studies showed that inhibition of PDGF results in reduced VSMC proliferation and migration after arterial balloon injury [79,80].

TGF-β, as opposed to PDGF, induces a contractile and non-proliferative VSMC phenotype. An absence of TGF-β results in severe congenital cardiovascular disease, mainly shown in structural defects in experimental animals [81]. In vitro treatment of VSMCs with TGF-β induces expression of *α*SMA, MHC and CNN1 [69,82,83].

Activated coagulation proteins also have been shown to affect VSMC phenotype via protease activated receptors (PARs) [84]. Both tumor necrosis factor-alpha (TNF-α) and angiotensin II have been shown to induce the contractile, as well as the synthetic, phenotype of VSMCs. It has been shown that angiotensin II affects VSMC phenotypic plasticity via the induction of ROS and decreased scavenging activity of nitrate reductase (NAD(P)H) oxidases [85] and subsequently induces aneurysm formation via oxidative stress [21,86].

#### 2.2.2. Extracellular Components

The ECM in which VSMCs are embedded also affects VSMC phenotype. VSMC phenotype is modulated via integrin receptors that are present on ECM proteins [87]. ECM consists of structural proteins, such as collagen, elastin and proteoglycans. For instance, the proteoglycan heparin promotes VSMC contractility. In vitro, heparin treatment of VSMCs induces a contractile phenotype and slows down its proliferation [69,88]. Collagen has pleiotropic effects on VSMC phenotype, depending on the type of collagen. Collagen type-I and fibronectin induce a synthetic VSMC phenotype [89,90] with proliferation [91,92]. On the contrary, collagen type-IV and laminin promote a contractile VSMC phenotype [89,93]. Intact elastin is associated with a contractile VSMC phenotype. The loss of elastin, by a genetic mutation, is associated with increased hypertrophy and hyperplasia of VSMCs [94,95]. Additionally, elastin remodeling, by the modulation of cathepsins and MMPs, promotes VSMC phenotypic switching that induces calcification [96,97].

Not only composition, but also the organization of structural fibers, determines VSMC phenotype. Culturing VSMCs in 3D compared to 2D increases contractile expression markers and induces TGF-β expression [98,99]. Culturing VSMCs on scaffold templates controls spatial organization and morphological response of VSMCs, indicating that VSMCs react to structural environmental changes to retain vessel function [100].

#### 2.2.3. Biophysical Factors

Hemodynamics at the vessel level (flow and transmural pressure) also affect VSMCs by determining tensile loading and active tone (e.g., by NO). Changes in vessel wall shear stress modulate EC NO release, and influence VSMCs via cell–cell interaction with ECs [101]. Using co-cultures of ECs and VSMCs, shear stress induces a synthetic VSMC phenotype by decreasing *α*-SMA, MYH and smtn expression [69]. However, 2D co-culture systems do not mimic the complex vessel wall architecture in vivo. Changes in blood pressure directly transfer into changes in ECM tensile stress, which, as determined by (local) matrix elastic behavior and VSMC stiffness, results in VSMC deformation (i.e., mechanical strain). Alterations in VSMC strain are understood to modulate cellular phenotype [33]. Mechanical stretching forces applied directly to VSMCs enhance the expression of ECM proteins, such as collagen and fibronectin [102]. Biophysical factors in the physiological range promote VSMCs to maintain or adopt a contractile phenotype, whereas pathological biophysical stimuli promotes VSMCs to switch towards a synthetic phenotype.

#### 2.2.4. Transcriptional Regulators

KLF4 is an important transcriptional regulator defining VSMC phenotype during development [103] and after vascular injury [104]. KLF4 expression is increased in lesions of ApoE^-/-^ mice on a Western type diet [105]. Additionally, there is increased binding of KLF4 to *α*SMA and SM22*α* promotors upon vascular injury in mouse carotid arteries [104]. The loss of KLF4 has favorable effects, inhibiting plaque pathogenesis and reducing plaque vulnerability [106]. KLF4 is critical in the regulation of phenotypic switching of VSMCs, both in vitro and in vascular injury models. KLF4 expression results in profound activation of pluripotency genes such as Oct4 and Sox2, indicating that VSMCs can reactivate its pluripotency network in response to vascular stress or damage [5]. Moreover, KLF4 activates > 800 pro-inflammatory VSMC genes, of which many are atherosclerosis relevant [106]. Additionally, KLF4 knock-out in an atherosclerosis animal model revealed the switch of VSMCs toward a synthetic phenotype, while suppressing the macrophage-like form [106].

## 3. Clinical Features of Vascular Remodeling

### 3.1. Hypertension

Hypertension is widely accepted as a risk factor for the development of CVD. However, hypertension was first considered to be a consequence of aging and seen as insurmountable [107]. Later, several studies revealed that hypertension was associated with increased cardiovascular mortality [108,109]. A meta-analysis on blood pressure and cardiovascular disease showed that a rise of 20 mmHg in systolic blood pressure (SBP) and of 10 mmHg in diastolic blood pressure (DBP) was associated with a more than two-fold increase in vascular mortality [110]. More recently, the CALIBER study revealed that patients with hypertension (defined as > 140/90 mm Hg or those receiving blood pressure-lowering drugs) have a lifetime risk for overall CVD of 63.3% at 30 years of age (compared to 46.1% in patients with normal blood pressure) and develop CVD 5 years earlier [111]. Decreasing the SBP by 20 mmHg was associated with a 39% reduction in cardiovascular (CV) events in the total group and in a 69% reduction in patients between 60 and 69 years of age [112]. Currently, much of the focus on the primary and secondary prevention of CVD revolves around control of blood pressure [112].

The systolic aspect of hypertension is quite strongly determined by age- and disease-related decreases in arterial compliance [9,111,112,113]. Therefore, arterial remodeling processes affecting medial elastic properties are directly relevant in risk profiling and as treatment targets.

#### 3.1.1. Cellular Components and Hypertension

High blood pressure is a multifactorial disorder in which genetic alterations, environmental factors and comorbidities interact [24,114]. ECs play an important role in the development of hypertension. In health, ECs are exposed to physiological shear stress, which is necessary to maintain proper functioning of the endothelium. The NO excreted by ECs regulates vascular tone, and thereby preserves ECM and VSMCs’ functioning [115]. In pathology, blood flow is more oscillatory, with higher peaks and blood stasis during diastole [116]. Turbulent flow and changes in shear stress, either high or low, affect EC function. Low shear stress areas are considered prone to developing atherosclerosis, inducing pro-inflammatory pathways and lead to dysregulation of the cytoskeleton and junctional proteins of ECs [115,117,118]. High shear stress induces morphological changes in ECs, as they align in the direction of the flow. Additionally, arterial outward remodeling occurs, leading to increased NO synthesis [119,120].

In hypertension, the amplitude and rate of elastic distension of arteries are increased, causing fatigue, damage and degradation of the vessel wall ECM [32]. Moreover, excessive VSMC strains induce phenotypic switching of these VSMCs to more synthetic types and, consequently, promotes arterial fibrosis, (further) compromising arterial compliance [121].

#### 3.1.2. Calcification and Hypertension

Several pathologies are associated with VSMC phenotype switching [12]. Synthetic VSMCs produce EVs [2,122,123], that have been found in both the intimal and medial layers of the vessel wall [124]. VSMCs are known to release EVs upon phenotypic switching towards synthetic or osteogenic phenotype. EVs derived from VSMCs share similarities with EVs from osteoblasts, having calcium-binding capacities and osteoblast-like ECM production [2,80]. Recently, it was shown that not all EVs promote calcification [125]. This implies that EV content is variable and dependent on VSMC-mediated biogenesis. A specific subclass of EVs are exosomes, which express tetraspanins (CD9, CD63 and CD81) and differ in expression pattern, and, hence, mineralizing capacity [125]. Furthermore, calcifying conditions in vitro increase the multi-vesicular body, forming enzyme sphingomyelin phosphodiesterase 3 (SMPD3) and subsequent exosome genesis [80]. Consequently, the inhibition of SMPD3 ablates the generation of exosomes and their subsequent calcification. Additionally, sortilin has been suggested as a key player in the VSMC-mediated calcifying EVs genesis and release. Sortilin is a sorting receptor that directs target proteins to a designated location via the secretory or endocytic compartment [126]. Recent findings have identified that sortilin promotes vascular calcification via the trafficking and loading of tissue-nonspecific alkaline phosphatase into EVs [127]. Moreover, sortilin co-localized with calcification in human calcified vessels [127]. Differences between mineralizing and non-mineralizing EVs eventually determine calcification of the ECM in proximity of VSMCs. Hence, better insight into EV composition, such as lipid content and RNA and protein profile, might provide new insights into the mechanism by which VSMC derived EVs contribute to vascular calcification and result in novel targets for treatment.

In response to calcified ECM, neighboring ECs and VSMCs react and induce the production of osteogenic factors, such as bone morphogenetic protein 2 (BMP-2) and 4 (BMP-4) [128,129]. Besides this expression of bone-associated proteins, the number of VSMCs decreases with age, and an increase of collagen-to-elastin ratio further increases the stiffening of the vessel wall [130].

In hypertension, calcium handling is disturbed, which is associated with an increased activation of L-type calcium channels and sensitivity of (hypertensive) patients toward calcium channel blockers [131]. Increased intracellular calcium induces the activation of receptors coupled to phospholipase C, leading to the generation of second messengers that trigger cytokines, ROS and miRNAs, and also cellular derived EVs [132,133]. Additionally, increased intracellular calcium activates the contractile machinery of VSMCs, leading to hyper-contractility. Excessive intracellular calcium rapidly disintegrates both mitochondria and structural components of VSMCs, and results in calcium depositions within elastic fibers [130]. Consequently, elasticity of the vasculature decreases, further contributing to increased blood pressure [24,134] and VSMC stiffening [135,136]. Specific calcium antagonists, blocking L-type voltage-dependent calcium channels, reduce calcium internalization and normalize blood pressure [137,138,139]. Of note, these calcium channel blockers also prevent calcification of the vasculature [140].

#### 3.1.3. Consequences of Vascular Calcification

Vascular calcification is associated with arterial stiffness [32,141]. In chronic kidney disease, medial calcification of large arteries is directly responsible for the increased stiffness (or reduced distensibility) [142,143]. The presence of vascular calcification (process) in the vessel wall contributes to ECM degradation and wall thickening [49,144]. Taken together, vascular calcification contributes directly and indirectly to arterial stiffening and, ultimately, hypertension [145,146]. Atherosclerosis has also been associated with arterial stiffness and hypertension [147,148]. Additionally, increased blood pressure and wall shear stress increase vulnerability and, consequently, atherosclerotic plaques become prone to rupture [30,149].

### 3.2. Atherosclerosis

Atherosclerosis is characterized by intimal plaque build-up and often referred to as a thickening of the vessel wall, ultimately narrowing the lumen of the vessel. Atherosclerosis is considered an inflammatory disease, starting with endothelial stress and subsequent monocyte infiltration [150,151]. Clinically, plaques may rupture, resulting in stroke or myocardial infarction [104]. Besides the classical view that systemic factors are causative for atherosclerosis, recent data suggest that local vascular processes also initiate atherogenesis [104]. Lineage tracing studies revealed that many of the cells present in atherosclerotic plaques are derived from VSMC precursors [30]. Additionally, it is now recognized that atherosclerotic plaque macrophages have lineage traces of VSMCs, indicating that cells in the intima originate from the vascular media, and not necessarily from the circulating mesenchymal cells that infiltrate the vessel wall at sites of injury [152].

Arterial remodeling is considered key to the development of atherosclerosis [148,153]. Arterial remodeling of the vessel wall can be initiated via many processes, including oxidative stress [2], proliferation [58], VSMC phenotype switching [154], cell infiltration, apoptosis and calcification [155,156,157].

### 3.3. Intimal and Medial Aspects of Vascular Calcification

Vascular calcification can be found at two anatomical locations, the medial and intimal layer. In the 1940s, post-mortem analysis of aortic specimens from CVD patients revealed that atherosclerosis is only present at vascular areas with either medial degradation or medial calcification [158,159,160,161]. Histological examination of aortas showed that medial calcification preceded atherosclerotic plaque build-up. Intimal calcification is seen as a hallmark of atherosclerosis and, clinically, is used as a marker for atherosclerotic burden [162]. For a long time, calcification has been considered as a passive degenerative detrimental process, not amendable for intervention. This view has recently changed, since statin treatment is associated with both increased plaque stability and increased plaque calcification [163,164,165]. However, the apparent protective role of calcification in stabilizing plaque is in contrast with the predictive value of calcification for CVD morbidity and mortality [160,164]. Moreover, medial calcification has been shown to change blood dynamics by inducing arterial remodeling processes and increasing arterial stiffness [166,167,168]. Medial calcification strongly correlates with the phenotypical switching of VSMCs and a rise in blood pressure. Macrocalcifications in patients who underwent a computed tomography (CT) scan were shown to have a three- to four-fold risk of developing fatal cardiovascular events [169]. Additionally, microcalcifications, which are precursors of macrocalcification, have been shown to induce plaque vulnerability [170].

#### Cellular Processes of Vascular Calcification

Specific stimuli, such as elevated calcium or phosphate levels, induce the switching of VSMCs to an osteogenic phenotypic where the cells acquire features of chondrocytes and osteoblasts [24]. Osteogenic VSMCs show an increased expression of osteogenic markers, such as alkaline phosphatase, BMP-2 and runt-related transcription factor 2 (Runx2), but show decreased calcification-inhibitor protein expression [171]. VSMCs exposed to elevated calcium levels display an intracellular calcium overload that may induce microcalcifications, eventually resulting in macrocalcification, which contributes to vascular stiffness and hypertension [172]. The process of calcification, culminating in either micro- or macrocalcification, compromises the structural integrity of the vessel wall and, hence, its functional properties.

Oxidative stress is an important player in the development of vascular calcification. Oxidative stress pathways in VSMCs have been linked to development aortic valve calcification [173]. Increased oxidative stress induces expression of Runx2, a key transcription factor associated with osteoblast differentiation [174]. VSMC-specific Runx2 knockout was shown to decrease vascular calcification and atherosclerosis development [175]. Additionally, increased H_2_O_2_ levels have been found in the close vicinity of calcification nucleation foci, together with an enhanced expression of oxidases [173]. High phosphate levels, present in chronic kidney disease, are associated with extent of vascular calcification [176]. In vitro, phosphate has been shown to induce mitochondrial-derived free radicals, by enhancing mitochondrial membrane potential [177]. Antioxidants inhibit phosphate-induced ROS production [177]. Increased oxidative stress is also found in diabetic patients [178], leading to an increased accumulation of advanced glycation end products (AGEs). AGEs have been shown to induce vascular calcification via increased oxidative stress of VSMC [179,180].

### 3.4. Fibrosis

Fibrosis of different organs contributes to up to 50% of worldwide mortality [181,182]. In the vasculature, myofibroblasts produce ECM proteins such as collagen and fibronectin [12]. Myofibroblasts are derived from MSC-like cells that reside in the vessel wall. The role of these MSC-like cells in the vessel wall is not fully understood, and is believed to play an important role in structural vessel repair. In vascular fibrosis, excessive collagen deposition leads to a reduced compliance of the vessel, which indicates increased arterial stiffness. Furthermore, high blood pressure induces structural changes including hypertrophy of the vessel wall [183,184] and cellular [185] and ECM remodeling [186].

Lineage tracing studies have shown that resident Gli1^+^ MSC-like cells are a major contributor of vascular fibrosis in arterial remodeling [187]. Acute vessel wall injury revealed that a significant number of Gli1^+^ cells were detected in the media and neointima, and that they express VSMC markers such as *α*SMA and CNN1. Additionally, some 50% of the newly formed VSMCs were derived from Gli^+^ progenitors, indicating the involvement of adventitial cells in arterial remodeling [188]. Likewise, angiotensin II induced hypertension was caused by an exuberant production of collagen in the adventitial layer [189]. VSMCs that migrate towards the adventitia start expressing stem cell markers Sca-1 and CD34 [186]. These “adventitial stem cells” change towards fibroblasts and deposit collagen and other ECM, contributing to adventitial fibrosis [190,191].

During atherosclerosis development, fate-tracing studies in mice have shown that MSC-like cells migrate into the media and neo-intima. Also, MSC-like Gli1^+^ cells were a major source of osteoblasts-like cells, significantly contributing to the process of vascular calcification in both vascular media and intima. The genetic ablation of Gli^+^ cells before injury dramatically reduced the severity of vascular calcification [192]. These data indicate that fibrotic processes are regulated by myofibroblasts and MSC-like cells and that these are important in the development of atherosclerosis and vascular calcification.

## 4. Assessing Vascular Remodeling and Disease

### 4.1. Pulse Wave Velocity (PWV), Distensibility Coefficient (DC) and Intima-Media Thickness (IMT)

Carotid-femoral PWV is considered the gold standard for assessing central arterial stiffness [193,194]. PWV, however, only measures vascular characteristics of the aorta and does not harbor information on any other vessels. Multi-slice CT, measuring calcification, reflects the atherosclerotic burden, and IMT measures atherosclerosis in the carotid arteries. Below, we describe in more detail measures of vascular remodeling and disease.

#### 4.1.1. Pulse Wave Velocity

Measuring arterial stiffness in the clinic is challenging. The methods that are available for measuring arterial stiffness are non-invasive vascular imaging, such as ultrasound and magnetic resonance imaging (MRI), and high-fidelity recording of the pulse wave, performed by tonometry or ultrasound Doppler. Functional measures that can be obtained are pulse wave velocity (PWV) and the distensibility coefficient (DC). In the clinic, carotid–femoral PWV (cfPWV) adds to existing risk scores in cardiovascular risk management [195], while arterial stiffening has been shown to be independently associated with cardiovascular risk and mortality [16]. Brachial-ankle PWV is also used as an alternative to cfPWV, correlating reasonably well [16].

The interpretation of arterial stiffness findings in human studies is subject to (1) the stiffness measure/method used and (2) the mechanistic or constitutive focus of the study. Constitutive properties that are often considered to interpret arterial stiffness findings are those related to the ECM or VSMCs [196,197].

#### 4.1.2. Distensibility Coefficient

Arterial stiffness can be determined by measuring local vessel diameter and distension by ultrasound or MRI [198]. The measured data can be used to calculate the local distensibility coefficient (DC) or compliance. PWV, as well as DC, are well known to be pressure dependent and require blood pressure adjustment [199]. Adjustment is valid when considering group analyses, but lacks applicability in the use of individual cases [32,200].

PWV is largely dependent on the elastic properties of the large arteries. Therefore, PWV should be interpreted with care, since it only reflects the stiffness of the aorta. In conduit arteries, the relative amount of VSMCs are limited and, thus, the contribution of ECM to arterial stiffening may be expected to be more pronounced. In peripheral, more muscular, arteries, age-related stiffening is not as straightforward [201,202]. PWV in the lower extremities can change significantly, which may explain why no evident relation between PWV and age has been reported [203]. Additionally, hypertrophy of VSMCs is commonly observed in hypertension [204], with the potential to (partially) determine PWV [205]. Thus, PWV is an important clinical marker for risk assessment, as it correlates with cardiovascular risk, mortality, and organ damage of the heart, kidney and brains [206,207].

Techniques for measuring PWV are developing. cfPWV is most commonly used for aortic stiffness, but is susceptible to local wave reflections and not suitable for obtaining information on carotid arteries [208]. Ultrasound ultrafast imaging techniques are being explored to directly measure the propagation of the pulse wave, with limited success for the common carotid artery [209]. Elastography is a promising technique, as it measures the propagation of shear waves in the tissue [210]. The propagation of the shear wave is directly proportional to the elastic modulus of the tissue [16]. PWV can also be measured using MRI, which also allows for segmentation. Its drawbacks are costs and long scanning times [33]. Currently, the available PWV toolbox can only be applied to large and medium-sized arteries, but techniques are developing for smaller vessels.

At present, the interpretation of stiffness measurements in terms of arterial wall constitutive properties remains a challenge [187]. Medial calcifications are known to co-localize with elastin fragmentation spots, but the mechanical contribution of the calcific deposits to vessel stiffness remains to be established; the elastin degradation alone could be sufficient to explain stiffening [211].

#### 4.1.3. Intima-Media Thickness

IMT is used as a measurement for assessing the thickness of the tunica intima and media (estimated from lumen-intima and media-adventitia echo complexes). IMT is often measured using ultrasound and can be readily obtained in superficial arteries. One aspect that is not measured by IMT is adventitial thickness is a significant contributor of pathology development [212]. Wall thickness data are important in determining arterial wall material stiffness and wall stress. But, IMT can also be used to detect atherosclerosis, and is used to follow progression of the disease [213,214]. IMT is shown to be an independent predictor of overall cardiovascular events, including myocardial infarction and stroke [215]. Multiple studies have shown that increases in IMT can be significantly delayed using effective cardiovascular drugs, such as statins [216,217]. However, further observational studies revealed that this delay did not result in reduced cardiovascular events [218].

To study large populations, non-invasive techniques such as IMT are preferred over invasive measurement tools. As such, IMT provides a reliable tool for assessing vessel wall macro-structural changes. However, some limitations for IMTs exist. There is a lack of standardized protocols for the use of IMT, and attempts to establish them have not reached consensus yet [219]. IMTs of certain segments, such as the common carotid artery, are related to hypertension, blood pressure and vascular hypertrophy [220], whereas the internal carotid artery correlates with atherosclerotic plaques [221,222]. Taken together, additional risk assessments, such as plaque presence and stenosis severity, will be complementary to the use of IMT alone. A novel technique to determine extra-media thickness may have great potential in combination with IMT to discriminate between the medial and adventitial layer which aids in risk assessment [223,224]. Moreover, other imaging techniques, such as coronary artery calcium (CAC) scanning, have been demonstrated to be superior for predicting CVD events [155,225].

### 4.2. Calcification Imaging by Computed Tomography

Calcification imaging in the clinic is mostly based on CT, which is an angiography technique based on X-ray radiation and makes use of computer processing to create a detailed 3D image. Calcification scores, such as the coronary artery calcium (CAC) score, can be derived from CT scans and, in recent decades, this has been used as a measure of atherosclerotic burden. The Agatson score is the most frequently used method to calculate CAC. The Agatston score is defined as a weighed sum of the calcification area, with density given as Hounsfield Units [226]. CAC represents an accurate measure to assess atherosclerotic burden [227,228]. Additionally, CAC correlates with traditional risk factors for atherosclerosis, such as hypertension, hypercholesterolemia and diabetes [227,228]. CAC also associates with cardiovascular morbidities and mortalities, including cardiac death and stroke [229].

CAC is regarded as the best clinical predictor for the accumulation and progression of vascular calcification over time [230,231,232]. CAC can be used to identify high-risk individuals that need immediate medical attention or intervention [233]. Interventions to slow down or even regress calcification, using the supplementation of vitamin K, are currently being conducted [234]. High dietary intake of vitamin K is associated with a lower risk of coronary heart disease [235,236,237] and supplementation with vitamin K significantly delays the progression of calcification [238]. Conversely, statin treatment is known for its beneficial cardiovascular effects, however, it is also shown to increase CAC score [239]. This indicates that type of calcification, as well as the volume or density of calcification, determines whether it is plaque-stabilizing or plaque-destabilizing. Indeed, CAC volume is positively and independently associated with CVD, whereas CAC density was inversely correlated with CVD risk [240].

Measuring CAC as a clinical marker for CVD has some limitations. Firstly, CAC is limited to coronary arteries, which do not represent the calcification of other vessels of the arterial system. CAC does not represent total plaque size, as histopathological studies showed that the area of calcium deposition was much smaller than total plaque volume [241,242]. Additionally, CAC score poorly correlated with luminal narrowing and severity [217,243,244]. Secondly, distinction between medial and intimal calcification is not possible with current resolutions of CT. Intimal and medial calcification are different pathologies, with different risk factors for developing CVD [245]. Additionally, microcalcification (< 15 μm) cannot be assessed, as the resolution of CT is some 200 μm [246].

Thirdly, evidence suggests that atherosclerotic plaques that have denser calcification have smaller lipid cores, with increased stability of plaques, resulting in lower CVD risk [247]. Greater CAC volume, however, is associated with a higher risk of CVD [248]. This indicates that future research should integrate density and volume into CAC scoring. Finally, CAC involves exposure to radiation, which limits the use of calcium scores as a follow-up on atherosclerosis progression [249].

### 4.3. ^18^F-NaF Positron Emission Tomography

^18^F-NaF PET scanning gained recent attention as a novel imaging modality for vascular calcification [250]. In bone, ^18^F-NaF is incorporated into newly formed hydroxyapatite crystals by exchanging fluoride with the hydroxyl groups [251]. In the vasculature, ^18^F-NaF PET imaging has been reported to identify active (micro)calcification with high specificity [252]. Importantly, CT and echocardiography cannot identify active vascular calcification, which is considered the main driver of disease progression [253]. Assessing the molecular bioactivity of calcification in atherosclerotic plaques is of additive value to identify vulnerable plaques [254,255]. Additionally, ex vivo histological studies showed that the resolution of ^18^F-NaF has the potential to discriminate between micro-and macrocalcification [254]. Taken together, ^18^F-NaF uptake provides complementary information to CT, as it allows for the detection of ongoing active calcification of plaques, which is a major improvement to the prediction of future cardiovascular risk. Limitations of ^18^F-NaF PET imaging are the relatively high costs and exposure to radiation, but, also, ^18^F-NaF PET imaging lacks the resolution to discriminate between medial and intimal calcification.

## 5. Conclusions

Arterial remodeling is defined as the adaptation of the vessel wall away from a normal homeostatic state towards pathologic development, triggered by chronic exposure to stress signals. Different arterial remodeling processes have been described, such as inflammation, oxidative stress, lipid accumulation, and degradation of the ECM. Many of these are regulated by VSMCs and affect vessel wall morphology and properties. VSMCs exhibit cellular plasticity and, upon stress stimuli, they may differentiate towards synthetic, macrophage-like and osteogenic VSMCs. As a consequence, VSMCs initiate and support remodeling processes involving ECM synthesis and cell proliferation, migration and contraction, which aid in maintaining tissue functionality. Disruption of VSMC phenotypic plasticity has many pathological consequences, including the development of hypertension, atherosclerosis, aneurysm formation, intimal and medial calcification, and vascular fibrosis, ultimately leading to increased CVD morbidity and mortality. Early detection of vascular remodeling is key in the prevention and prognosis of CVD. Clinically, assessment of calcification-mediated remodeling remains difficult. Assessment tools include measurements of pulse wave velocity, distensibility coefficient and intima-media thickness. These techniques may capture changes in dynamic and geometric vessel wall properties, but cannot directly monitor the underlying arterial remodeling processes. The addition of techniques, such as ^18^F-NaF PET and CT imaging may provide further mechanistic insight into vascular calcification development and other VSMC-mediated processes.

Combining expertise in vascular biology, clinical vascular expertise and multi-scale/multi-modality imaging should advance the understanding and identification of critical determinants of vascular remodeling, ultimately resulting in tailor-made diagnoses and treatments.

## Figures and Tables

**Figure 1 ijms-20-05694-f001:**
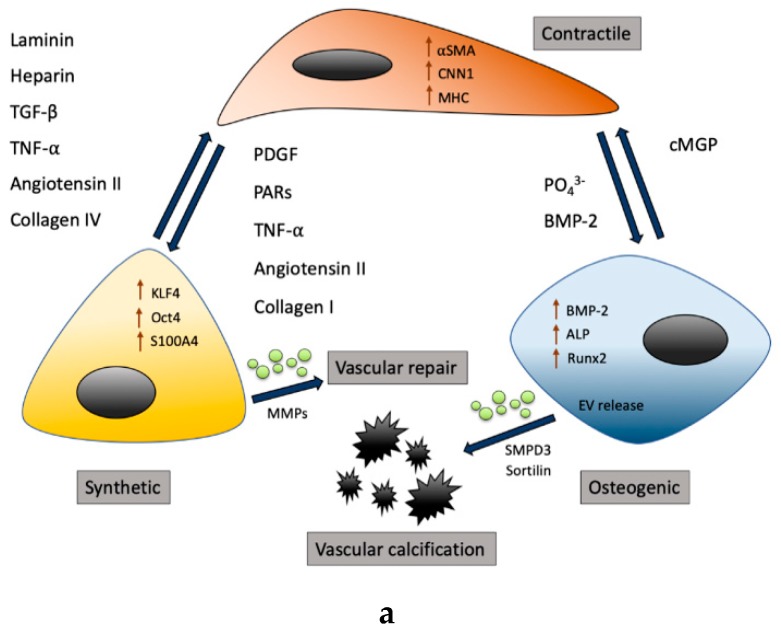
(**a**) Vascular smooth muscle cell (VMSC) phenotypic switching. Under physiological conditions, VSMCs display a contractile phenotype that regulates vessel structure and function. Under stress, arterial remodeling will occur, leading to VSMC phenotypic switching. Factors inducing the synthetic phenotype include platelet derived growth factors (PDGF), protease-activated receptors (PARs) and tumor necrosis factor-alpha (TNF-α). Synthetic VSMCs initiate vessel repair and can switch back to the contractile phenotype, driven by factors such as heparin or laminin. The osteogenic phenotype can be induced by prolonged exposure to BMP-2 or high phosphate. Osteogenic VSMCs shed extracellular vesicles that promote vascular calcification. Panel (**b**). Pathways affecting vascular calcification. Vascular calcification is an active process which can be initiated by several pathways, including: 1. biochemical factors, 2. physical factors, 3. vascular calcification (VC) inhibitors and 4. ECM factors. Biochemical factors, such as raised calcium and phosphate levels and PDGF, are stress molecules that induce VSMCs to switch towards a synthetic or osteogenic phenotype, including an increased release of extracellular vesicles. The ECM of the vessel wall directly influences VSMCs. Changes in collagen and elastin content cause VSMCs to change morphology and phenotype. VSMCs in turn, produce MMPs that induce structural changes in the vessel wall by rearranging collagen and elastin, promoting the migration and proliferation of VSMCs and other cell types. Physical factors such as shear stress and tensile stress affect ECs’ NO release, which influences the VSMC phenotype. Shear-induced stress induces a synthetic phenotype by decreasing a-SMA, MYH and smtn expression. Tensile stretch induces VSMCs to produce ECM proteins, such as collagen and fibronectin promoting vessel fibrosis. VC inhibitors, such as OPN, MGP and PP_1,_ affect the VSMC phenotype by inducing changes in RNA expression patterns. Taken together, all stimuli play a part in an orchestrated VSMC response, ultimately promoting vascular calcification.

**Figure 2 ijms-20-05694-f002:**
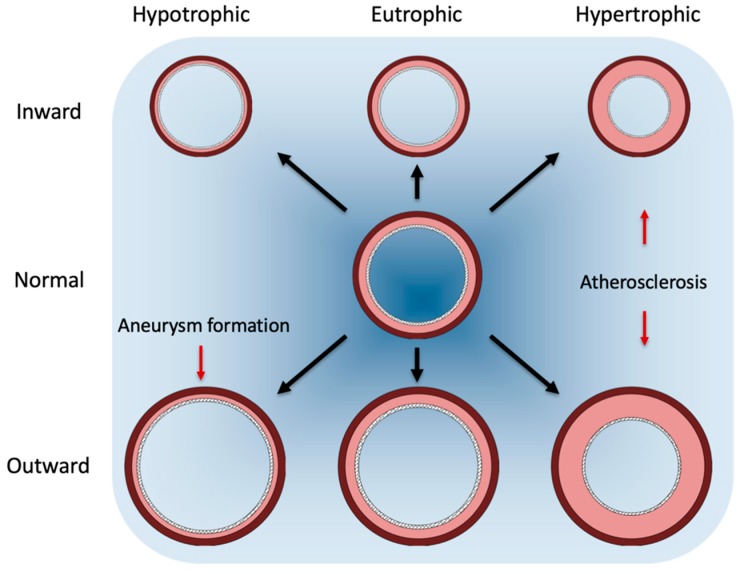
Vascular remodeling types. There are several types of arterial remodeling: hypotrophic, eutrophic and hypertrophic. Additionally, remodeling can be inward and outward. Hypotrophic remodeling results in a thinner vessel wall, which can be both inward and outward. In both cases, the wall-to-lumen ratio decreases. Hypertrophic remodeling results in the thickening of the vessel wall, that can also be inward and outward. The thickening of the vessel wall results in an increased wall-to-lumen ratio. In the eutrophic situation, wall-to-lumen ratios do not differ, but the size of the vessel can change. Atherosclerosis is characterized by an increased wall-to-lumen ratio, with both thickening of the media and intima and, therefore, is classified as inward or outward hypertrophic remodeling. Aneurysm formation is characterized by an increase in vessel diameter with a thinning of the vessel wall (outward hypotrophic remodeling). Inward remodeling is less frequent and is more associated with muscular peripheral arteries, reflecting sustained vasoconstriction of vessels.

**Figure 3 ijms-20-05694-f003:**
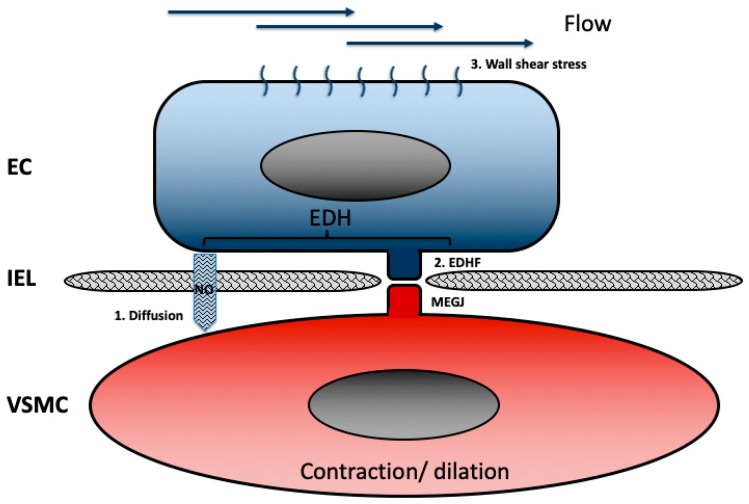
Endothelial cell–smooth muscle cell communication. Endothelial cells (ECs) communicate with vascular smooth muscle cells (VSMCs) in two ways, called endothelial derived hyperpolarization (EDH). EDH has two major pathways: 1. Diffusion of nitric oxide (NO), which is produced by ECs and diffuses through the internal elastic lamina (IEL) to induce the relaxation of VSMCs. 2. Signaling through myoendothelial gap junctions (MEGJ) that connect ECs directly to VSMCs and cross the IEL. This direct communication between ECs and VSMC via MEGJ occurs via so-called endothelial derived hyperpolarizing factors (EDHF). 3. Major factors that stimulate EC NO secretion are flow or shear-stress alterations. Differences in wall shear rates are known to induce endothelial-derived changes in VSMC phenotype.

**Table 1 ijms-20-05694-t001:** Proteins that define VSMC phenotype.

Protein	Gene Name	Abbreviation	Expression	
			Contractile	Synthetic
Alpha smooth muscle actin	*ACTA-2*	⍺-SMA	+	+
Smooth muscle myosin heavy chain	*MYH11*	SMMHC	+	−
Smooth muscle 22 alpha	*TAGLN*	SM22-⍺	+	−
Smoothelin	*SMTN*	Smtn	+	−
Calponin	*CNN-1*	CNN-1	+	−
Tumor necrosis factor alpha	*TNFA*	TNF-⍺		+
S100 calcium binding protein A4	*S100A4*	S100A4	−	+
Monocyte chemoattractant protein 1	*CCL2*	MCP-1	−	+

+, positive effect; −, negative effect.

**Table 2 ijms-20-05694-t002:** Factors involved in VSMC phenotype switching.

Factors Involved in VSMC Phenotype Switching	Phenotype	
Biochemical compounds	Contractile	Synthetic
PDGF	−	+
TGF-β	+	−
PARs	−	+
TNF-⍺	+	+
Angiotensin II	+	+
Extracellular matrix components		
Integrin: ⍺1β1, ⍺7β1, ⍺8β1	+	−
Integrin: ⍺2β1, ⍺5β1, ⍺vβ3	−	+
Collagen type I	−	+
Collagen type IV	+	-
Elastin	+	-
Heparin	+	-
Fibronectin	−	+
Laminin	+	−
Physical factors		
Tensile stress	−	+
Shear stress	−	+
Transcription		
KLF4	−	+
Oct4	−	+

Abbreviations: PDGF, platelet derived growth factor; TGF-β, transforming growth factor beta; PARs Protease-activated receptors; TNF-⍺, tumor necrosis factor alpha; KLF4, krüppel-like factor 4; Oct4, octamer-binding transcription factor 4. +, positive effect; −, negative effect.

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
