# Peer review of "The Role of Vascular Smooth Muscle Cells in Arterial Remodeling: Focus on Calcification-Related Processes"

_ijms, 2019, doi:10.3390/ijms20225694_

Round 1
Reviewer 1 Report
This is a well-written paper with comprehensive review of mechanisms of calcification-related vascular remodeling.
I have only one concern.
The figure one needs to be redone. I suggest that the figure should be illustrated in a way with components of EC,VSMC, ECM according to anatomic alignment. Some texts can not be visualized clearly.
Author Response
We thank the reviewer for critically reading our manuscript and providing his/ her suggestions that will improve our review.
Reviewer #1:
The figure one needs to be redone. I suggest that the figure should be illustrated in a way with components of EC, VSMC, ECM according to anatomic alignment. Some texts cannot be visualized clearly.Response: Figure 1 has been adjusted and includes anatomical alignment and all vascular structures. Texts are adjusted to make them better visual.
Reviewer 2 Report
In the present review the authors highlight the role of vascular smooth muscle cells (VSMC) in regulating remodeling processes of the vessel wall. In particular they focalize the attention on processes that govern arterial stiffening, atherosclerosis and calcification also describing clinically applicable methodologies to assess arterial remodeling. In my opinion the manuscript needs of some improvements before publication.
Listed below some specific comments:
In the last decade, the role of immune system has garnered attention by vascular biology. The authors should describe the important role of immunity in vascular remodeling.In order to make the review more understandable for the reader, the authors should elaborate a cartoon reporting the main molecular mechanisms involved in the VSMC remodeling. This would give more impact to the review.
Conclusion section should be expanded.
Exosome, a newly identified natural nanocarrier and intercellular messenger, plays a pivotal role in regulating cell-to-cell communication. Recently, it has been reported its implication in vascular calcification and remodeling. This point should be discussed.
ROS, and in particular mitochondrial reactive oxygen species exert an important role on the regulation of phosphate-induced vascular calcification both in vitro and in vivo. Please discuss.
Sortilin has an important role in calcification process in CVDs. The authors should at least discuss its vascular effects.
Author Response
We thank the reviewer for critically reading our manuscript and providing his/ her suggestions that will improve our review.
Reviewer #2:
In the last decade, the role of immune system has garnered attention by vascular biology. The authors should describe the important role of immunity in vascular remodeling.Response: we do agree that the role on the immune system is important in relation to vascular biology, we have therefore added information about the role of the immune system to aspects of vascular remodeling (section 2.1.1) anther part was added about the role of inflammation in transcription (section 2.2.4):
“2.1.1 Inflammation and arterial remodeling
Inflammatory processes play a significant role in arterial remodeling and are linked to the development of atherosclerosis. Chronic low-grade inflammation is a key driver of arterial aging. VSMC phenotypic switching and ECM biochemical changes activate inflammatory pathways and oxidative stress signaling [54]. Cytokines determine processes such as endothelial dysfunction, renin angiotensin activation and metalloproteinases release [55,56]. VSMCs are involvend in local vessel wall inflammatory functions.
Proliferation. VSMCs have low turnover in healthy vessels but increase proliferation upon vascular injury to initiate repair [57]. Increased expression of aging-genes p16INK4a and p21 [58] have been linked to VSMC senescence and are present in atherosclerotic plaques indicating lower proliferation and thus decreased repair capacity [59]. Cell death. Cell death is linked to vascular disease but absolute rates are difficult determine. VSMC apoptosis in atherosclerosis has been linked to plaque vulnerability features [60] and chronic VSMC apoptosis has shown to promote atherogenesis and plaque progression [61]. In vascular aging and medial remodeling, VSMCs cell death does not cause significant inflammation, due to efficient IL-1 mediated clearance of apoptotic bodies by VSMCs [62]. Platelets and extracellular vesicles (EVs). Platelets have shown to affect VSMC inflammatory phenotypes and injury responses [63]. Additionally, platelet EVs induce an inflammatrory repsonse in VSMC in vitro [64]. Platelet factor 4 (PF4) has a central role in the stimulation of VSMC-mediated cytokine release, which is primarily affected by increased krüppel-like factor 4 (KLF-4) transcription. Experimental atherosclerosis models indicated that PF4 is proatherogenic and found to penetrate deep into the vessel wall, underlining the importance of platelets aside from their effect on the endothelium [65].”
“Moreover, KLF4 activates >800 pro-inflammatory VSMC genes, in which many are atherosclerosis relevant [107]. Additionally, KLF4 knock-out in an atherosclerosis animal model revealed the switch of VSMCs toward a synthetic phenotype while suppressing the macrophage-like form [107].”
In order to make the review more understandable for the reader, the authors should elaborate a cartoon reporting the main molecular mechanisms involved in the VSMC remodeling. This would give more impact to the review.
Response: we agree that there was no figure that displayed the function of VSMCs in arterial remodeling processes. Therefore we adjusted figure 1 in which we now visualize the molecular mechanisms of VSMCs phenotype switching and connect it to remodeling.
Conclusion section should be expanded.
Response: based on this remark we extended the conclusion section with information about arterial remodeling. The conclusion as it reads now:
“5. Conclusion
Arterial remodeling is defined as the adaptation of the vessel wall away from a normal homeostatic state towards pathologic development triggered by chronic exposure to stress signals. Different arterial remodeling processes have been described, such as inflammation, oxidative stress, lipid accumulation, and degradation of the ECM. Many of these are regulated by VSMCs and affect vessel wall morphology and properties. VSMCs exhibit cellular plasticity and upon stress stimuli they may differentiate towards synthetic, macrophage-like and osteogenic VSMCs. As a concequence, VSMCs initiate and support remodeling processes involving ECM synthesis and cell proliferation, migration and contraction, which aid in maintaining tissue functionality. Disruption of VSMC phenotypic plasticity has many pathological consequences, including the development of hypertension, atherosclerosis, aneurysm formation, intimal and medial calcification, and vascular fibrosis, ultimately leading to increased CVD morbidity and mortality. Early detection of vascular remodeling is key in the prevention and prognosis of CVD. Clinically, assessment of calcification-mediated remodeling remains difficult. Assessment tool include measurements of pulse wave velocity, distensibility coefficient and intima-media thickness. These techniques may capture changes in dynamic and geometric vessel wall properties, but cannot directly monitor the underlying arterial remodeling processes. Addition of techniques such as 18F-NaF PET and CT imaging may provide further mechanistic insight into vascular calcification development and other VSMC-mediated processes.
Combining expertise in vascular biology, clinical vascular expertise and multi-scale/ multi-modality imaging should advance the understanding and identification of critical determinants of vascular remodeling ultimately resulting in tailor-made diagnosis and treatment.”
Exosome, a newly identified natural nanocarrier and intercellular messenger, plays a pivotal role in regulating cell-to-cell communication. Recently, it has been reported its implication in vascular calcification and remodeling. This point should be discussed.
Response: we agree with the reviewer that exosomes play a pivotal role in the regulation of vascular calcification. We see that our part about extracellular vesicles was open for interpretation and changed the structure of this part in which we included the term exosomes (section 3.1.2, first paragraph):
“3.1.2 Calcification and hypertension
Several pathologies are associated with VSMC phenotype switching [12]. Synthetic VSMCs produce EVs [2,123,124], that have been found in both the intimal and medial layers of the vessel wall [125]. VSMCs are known to release EVs upon phenotypic switching towards synthetic or osteogenic phenotype. EVs derived from VSMCs share similarities with EVs from osteoblasts, having calcium binding capacity and osteoblast-like ECM production [2,81]. Recently, it was shown that not all EVs promote calcification [126]. This implies that EV content is variable and dependent on VSMC-mediated biogenesis. A specific subclass of EVs are exosome which express tetraspanins (CD9, CD63 and CD81) and differ in expression pattern, and hence mineralizing capacity [126]. Furthermore, calcifying conditions in vitro increase the multi-vesicular body forming enzyme sphingomyelin phosphodiesterase 3 (SMPD3) and subsequent exosome genesis [81]. Consequently, inhibition of SMPD3 ablates generation of exosomes and subsequent calcification. Additionally, sortilin has been suggested as a key player in VSMC-mediated calcifying EVs genesis and release. Sortilin is a sorting receptor that directs target proteins to a designated location via the secretory or endocytic compartment [127]. Recent findings have identified that sortilin promotes vascular calcification via trafficking and loading of tissue-nonspecific alkaline phosphatase into EVs [128]. Moreover, sortilin co-localized with calcification in human calcified vessels [128]. Differences between mineralizing and non-mineralizing EVs eventually determine calcification of the ECM in proximity of VSMCs. Hence, better insight in EV composition such as lipid content and RNA and protein profile might provide new insights into the mechanism by which VSMC derived EVs contribute to vascular calcification and result in novel targets for treatment.”
ROS, and in particular mitochondrial reactive oxygen species exert an important role on the regulation of phosphate-induced vascular calcification both in vitro and in vivo. Please discuss.
Response: we thank the reviewer for pointing out the importance of ROS and vascular calcification. Therefore, we included a paragraph that describes the role of oxidative stress in vascular calcification (section 3.3.1, second paragraph):
“Oxidative stress is an important player in the development of vascular calcification. Oxidative stress pathways in VSMCs have been linked to development aortic valve calcification [174]. Increased oxidative stress induces expression of Runx2, a key transcription factor associated with osteoblast differentiation [175]. VSMC specific Runx2 knockout was shown to decrease vascular calcification and atherosclerosis development [176]. Additionally, increased H2O2 levels have been found in the close vicinity of calcification nucleation foci together with enhanced expression of oxidases [174]. High phosphate levels, present in chronic kidney disease, are associated with extent of vascular calcification [177]. In vitro, phosphate has been shown to induce mitochondrial derived free radicals by enhancing mitochondrial membrane potential [178]. Antioxidants inhibit phosphate-induced ROS production [178]. Increased oxidative stress is also found in diabetic patients [179], leading to increased accumulation of advanced glycation end products (AGEs). AGEs have been shown to induce vascular calcification via increased oxidative stress of VSMC [180,181].”
Sortilin has an important role in calcification process in CVDs. The authors should at least discuss its vascular effects.
Response: we agree that sortilin is important in the signaling of extracellular vesicles, and we described its role with regard to EV formation. This part can be found at the same location as the part about exosomes (see point 4).
Reviewer 3 Report
This is a very comprehensive review to introduce the calcification-related processes of VSMCs. Including the general introduction, molecular biological mechanism, clinical assessment, and diagnoses. Considered accepted with minor typo and spell check required as below.
Figure 1 paragraph (line81 ~line95) should be justify full.
Line 49, line 135, and line 139: “ageing” should be “aging”;
Ling 52: “pin-point” should be “pin point”;
Line 55: “unravelling” should be “to unravel”;
Line 65: “synthesise” should be “synthesize”;
Figure 1 paragraph (line81 ~line95) should be justify full;
Line 82: “Under physiological conditions VSMCs…” should be “Under physiological conditions, VSMCs…”;
Line 91: “shear induced” should be “shear-induced”;
Line 93: “produces” should be “produce”;
Line 98: “adaption” should be “an adaption”;
Line 100: “with respectively” should be “with respective”;
Line 103 and 105: “hypertropic” should be “hypertrophic”;
Line 112: “thickening” should be “the thickening”;
Line 114, line116, and line 324: “characterised” should be “characterized”;
Line 122: “an increased” should be “increased”;
Line 123: “recruitment” should be “the recruitment”;
Line 125 and line 220: “fibres” should be “fibers”;
Line 142: “content” should be “the content”;
Line 163: “…cell communication.Endothelial…” should be “…cell communication. Endothelial…”
Line 213: “type” should be “the type”;
Line 220 and line 222: “organisation” should be “organization”;
Line 232: “behaviour” should be “behavior”;
Line 237 “promote” should be “promotes”;
Line 243: “favourable” should be “favorable”;
Line 250: “risk” should be “a risk”;
Line 257: “life-time” should be “lifetime”;
Line 265: “target” should be “targets”;
Line 278: “is” should be “are”;
Line 285: “layer” should be “layers”;
Line 290, line293, and line195: “mineralising” should be “mineralizing”;
Line 298: “neighbouring” should be “neighboring”;
Line 301: “stiffening” should be “the stiffening”;
Line 305: “generation” should be “the generation”;
Line 312: “internalisation” should be “internalization”, “normalise” should be “normalize”;
Line 331: “recognise” should be “recognize”;
Line 327: “beside” should be “besides”;
Line 334: “key in” should be “key to”;
Line 344: “marker” should be “a marker”;
Line 347 and line 479: “stabilising” should be “stabilizing”;
Line 329: “tracking” should be “tracing”, should be uniformed in the whole context.
Line 366: “for up” should be “to up”;
Line 378: “involvement” should be “an involvement”;
Line 386: “severity” should be “the severity”;
Line 393: “harbour” should be “harbor”;
Line 405: “alternative” should be “an alternative”;
Line 422: “potential” should be “the potential”;
Line 433: “medium sized” should be “medium-sized”;
Line439: “as measure” should be “as a measurement”;
Line 451: “standardlised” should be “standardized”;
Line 464: “in the last decades this is” should be “in the last decades, this is”;
Line 465: “calculate of CAC” should be “calculate CAC”;
Line 471: “regarded” should be “regarded as”;
Line 475: “lower” should be “a lower”;
Line 476: “progression” should be “the progression”;
Line 477: “however it is” should be “however, it is”;
Line 478: “type” should be “the type”;
Line 482: “as” should be “as a”;
Line 484: “area” should be “the area”;
Line 494: “which limit use of calcium scores as follow-up” should be “which limit the use of calcium scores as a follow-up”;
Line 496: “attention as novel” should be “attention as a novel”;
Line 498: “exchange” should be “exchanging”;
Line 506: “to prediction of” should be “to the prediction of”;
Line 508: “lacks resolution” should be “lacks the resolution”;
Line 511: “specialised” should be “specialized”;
Line 514: “including development” should be “including the development”;
Line 518: “insight in cardiovascular” should be “insight into cardiovascular”;
Line 521: “may results” should be “may result”;
Line 528 abbreviations should include “VSMC”; “Smtn” should be “smtn”
Some of the corrections above might not be needed, please note.
Author Response
We thank the reviewer for critically reading our manuscript and providing his/ her suggestions that will improve our review.
Reviewer #3
Considered accepted with minor typo and spell check required as below.Response: We are thankful to the reviewer that our review was considered accepted. We changed all the listed spelling errors:
Figure 1 paragraph (line81 ~line95) should be justify full.
Line 49, line 135, and line 139: “ageing” should be “aging”;
Ling 52: “pin-point” should be “pin point”;
Line 55: “unravelling” should be “to unravel”;
Line 65: “synthesise” should be “synthesize”;
Figure 1 paragraph (line81 ~line95) should be justify full;
Line 82: “Under physiological conditions VSMCs…” should be “Under physiological conditions, VSMCs…”;
Line 91: “shear induced” should be “shear-induced”;
Line 93: “produces” should be “produce”;
Line 98: “adaption” should be “an adaption”;
Line 100: “with respectively” should be “with respective”;
Line 103 and 105: “hypertropic” should be “hypertrophic”;
Line 112: “thickening” should be “the thickening”;
Line 114, line116, and line 324: “characterised” should be “characterized”;
Line 122: “an increased” should be “increased”;
Line 123: “recruitment” should be “the recruitment”;
Line 125 and line 220: “fibres” should be “fibers”;
Line 142: “content” should be “the content”;
Line 163: “…cell communication.Endothelial…” should be “…cell communication. Endothelial…”
Line 213: “type” should be “the type”;
Line 220 and line 222: “organisation” should be “organization”;
Line 232: “behaviour” should be “behavior”;
Line 237 “promote” should be “promotes”;
Line 243: “favourable” should be “favorable”;
Line 250: “risk” should be “a risk”;
Line 257: “life-time” should be “lifetime”;
Line 265: “target” should be “targets”;
Line 278: “is” should be “are”;
Line 285: “layer” should be “layers”;
Line 290, line293, and line195: “mineralising” should be “mineralizing”;
Line 298: “neighbouring” should be “neighboring”;
Line 301: “stiffening” should be “the stiffening”;
Line 305: “generation” should be “the generation”;
Line 312: “internalisation” should be “internalization”, “normalise” should be “normalize”;
Line 331: “recognise” should be “recognize”;
Line 327: “beside” should be “besides”;
Line 334: “key in” should be “key to”;
Line 344: “marker” should be “a marker”;
Line 347 and line 479: “stabilising” should be “stabilizing”;
Line 329: “tracking” should be “tracing”, should be uniformed in the whole context.
Line 366: “for up” should be “to up”;
Line 378: “involvement” should be “an involvement”;
Line 386: “severity” should be “the severity”;
Line 393: “harbour” should be “harbor”;
Line 405: “alternative” should be “an alternative”;
Line 422: “potential” should be “the potential”;
Line 433: “medium sized” should be “medium-sized”;
Line439: “as measure” should be “as a measurement”;
Line 451: “standardlised” should be “standardized”;
Line 464: “in the last decades this is” should be “in the last decades, this is”;
Line 465: “calculate of CAC” should be “calculate CAC”;
Line 471: “regarded” should be “regarded as”;
Line 475: “lower” should be “a lower”;
Line 476: “progression” should be “the progression”;
Line 477: “however it is” should be “however, it is”;
Line 478: “type” should be “the type”;
Line 482: “as” should be “as a”;
Line 484: “area” should be “the area”;
Line 494: “which limit use of calcium scores as follow-up” should be “which limit the use of calcium scores as a follow-up”;
Line 496: “attention as novel” should be “attention as a novel”;
Line 498: “exchange” should be “exchanging”;
Line 506: “to prediction of” should be “to the prediction of”;
Line 508: “lacks resolution” should be “lacks the resolution”;
Line 511: “specialised” should be “specialized”;
Line 514: “including development” should be “including the development”;
Line 518: “insight in cardiovascular” should be “insight into cardiovascular”;
Line 521: “may results” should be “may result”;
Line 528 abbreviations should include “VSMC”; “Smtn” should be “smtn”
Round 2
Reviewer 2 Report
The authors have addressed all issues raised by this reviewer.